# Oral Microbiota in Children and Adolescents with Type 1 Diabetes Mellitus: Novel Insights into the Pathogenesis of Dental and Periodontal Disease

**DOI:** 10.3390/microorganisms11030668

**Published:** 2023-03-06

**Authors:** Maria Carelli, Alice Maguolo, Chiara Zusi, Francesca Olivieri, Federica Emiliani, Gelinda De Grandi, Ilaria Unali, Nicoletta Zerman, Caterina Signoretto, Claudio Maffeis

**Affiliations:** 1Department of Diagnostic and Public Health, Microbiology Section, University of Verona, 37134 Verona, Italy; 2School of Health Statistics and Biometrics, University of Verona, Strada Le Grazie 8, 37134 Verona, Italy; 3Section of Pediatric Diabetes and Metabolism, Department of Surgery, Dentistry, Pediatrics and Gynecology, University of Verona, 37126 Verona, Italy; 4Department of Surgery, Dentistry, Paediatrics and Gynecology, University of Verona, 37134 Verona, Italy

**Keywords:** type 1 diabetes, glycemic control, dental physiology, dysbiosis, cariogenic agents, microbial consortia, dental caries, periodontal disease, host–pathogen interaction, diabetes-related complication, oral hygiene, dental device homecare, primary prevention

## Abstract

The oral microbiota can be influenced by multiple factors, but only a few studies have focused on the role of glycemic control in determining early alterations of oral microbiota and their association with pathogenesis of both periodontitis and caries. The aim of this study is to evaluate the interplay between bacteria composition, oral hygiene, and glycemic control in a cohort of children with T1D. A total of 89 T1D children were enrolled (62% males, mean age: 12.6 ± 2.2 years). Physical and clinical characteristics, glucometabolic parameters, insulin treatment, and oral hygiene habits data were collected. Microbiological analysis was performed from saliva samples. A high prevalence of cariogenic and periodontopathogens bacteria in our cohort was detected. In particular, in all subjects *Actinomyces* spp., *Aggregatibacter actinomycetemcomitans*, *Prevotella intermedia*, and *Lactobacillus* spp. were isolated. *S. mutans* was found in about half of the analyzed sample (49.4%), in particular in patients with imbalance values of glycemic control. Moreover, a higher presence of both *S. mutans* and *Veillonella* spp. was detected in subjects with poorer glycemic control, in terms of HbA1c, %TIR and %TAR, even adjusting for age, sex, and hygiene habits as covariates. Virtuous oral hygiene habits, such as frequency of toothbrush changes and professional oral hygiene, negatively correlated with the simultaneous presence of *Tannerella forsythia*, *Treponema denticola*, and *Porphyromonas gingivalis*, red complex bacteria. Our study shows it is crucial to pay attention to glycemic control and regular oral hygiene to prevent the establishment of an oral microbiota predisposing to dental and periodontal pathology in subjects with T1D since childhood.

## 1. Introduction

Type 1 diabetes (T1D) is a complex autoimmune disease caused by the destruction of pancreatic beta cells that leads to both acute and chronic complications [1]. An inadequate glycemic control is a major risk factor for the development of chronic complications, including cardiovascular disease, peripheral vascular disease, retinopathy, nephropathy, and neuropathy [2,3,4,5]. Periodontal disease (PD) is classified as the “sixth complication” of diabetes [6]. This chronic inflammation of periodontal tissues is characterized by the progressive destruction of the supporting structures of the teeth induced by a state of dysbiosis promoting host inflammatory response [7,8]. According to the new classification of PD, although there is sufficient evidence to believe that PD observed in the context of systemic disease that severely impair the immune response should be considered a periodontal manifestation of systemic disease, there is currently insufficient evidence to sustain that PD observed in poorly controlled diabetes is characterized by a unique pathophysiology [9]. The incidence of PD in patients with T1D is higher than in the healthy population and is significantly associated with a longer duration of diabetes and poor glycemic control [10,11]. Indeed, the latter together with changes in host response and differences in the composition of the oral microbiota are suggested as determinants of increased susceptibility of T1D patients to the development of PD and caries [12]. Two meta-analyses have recently confirmed the association between diabetes and periodontal disease, indicating a positive, bidirectional association between these two disorders [13,14]. On one hand, diabetes increases the risk and severity of inflammatory PD; on the other, periodontitis can trigger inflammatory host immune responses locally and systemically, affecting glucometabolic control in patients with T1D [15,16,17]. This relationship impacts on early onset of gingival disease and increased periodontal disease even in children and adolescents with T1D [10,11].

Different combinations of bacterial species are involved in periodontitis, such as the concomitant presence of *Aggregatibacter actinomycetemcomitans* and *Prevotella intermedia* in saliva [18]. In addition, the “red complex”, consisting of three strictly anaerobic bacteria, i.e., *Porphyromonas gingivalis*, *Tannerella forsythia*, and *Treponema denticola*, is associated with severe forms of periodontal disease [19,20].

Moreover, a relationship between T1D and an increased risk of dental caries has been suggested. This link is influenced by diabetes-induced changes in saliva composition and levels of glycemic control [21,22]. As regards microbial composition, *Streptococcus mutans* and *Lactobacillus* are the most cariogenic bacteria because of their ability to survive in an acidic environment and form biofilm [23]. A molecular study found that the simultaneous presence of *Veillonella* spp. and *Streptococcus* spp. can promote the development and progression of dental caries [24,25].

In the general population, oral microbiome is suspected to be affected by several variables including host genetics, geography, age, cohabitation, and familial relationship. In particular, the salivary microbiota of youths, aged 3 to 18 years, is still maturing [26,27], while, as they age, the composition of the oral microbiome changes and periodontal pathogens increase in abundance, leading to increased susceptibility to oral disease [28]. Generally, children and adolescents have an oral microbiota characterized by bacteria that are protective against periodontal disease and caries, whereas people > 50 years of age show changes in the microenvironment that lead to an increase in certain bacterial species that predispose to periodontitis. For example, the study by Rodenburg et al. showed that the prevalence of periodontopathogens such as *Porphyromonas gingivalis* in individuals with periodontitis increases with age [29]. Proper education of the patient with T1D in hygiene maintenance with professional oral hygiene sessions with special tools could positively affect the maintenance of oral and dental health by preventing oral dysbiosis [30].

However, the oral microbiota composition may be influenced by additional multiple modifiable factors: dietary habits, oral hygiene, and use of drugs or antibiotics. Currently, few studies have focused on the role of glycemic control, analyzing different CGM metrics, in determining alterations in the oral microbiota of subjects with T1D, and, to date, its association with pathogenesis of both periodontitis and caries remains controversial [31,32,33]. A few data are available in children and adolescents with T1D. Clarifying the relationship between the oral microbiota and the dental and metabolic health of T1D individuals from an early age is crucial in order to develop novel, effective, preventative, and therapeutic strategies.

Therefore, the aim of this study was to assess the presence of cariogenic and periodontopathogenic bacteria through saliva sample analysis and evaluate their potential roles in the interplay with oral hygiene and glycemic control in a cohort of children and adolescents with T1D.

## 2. Materials and Methods

### 2.1. Study Population

Eighty-nine children and adolescents with T1D (age: 12.6 ± 2.2 years, 55 boys) were consecutively recruited at the Regional Center for Pediatric Diabetes of the University Hospital, Verona (Italy) during a follow-up visit between December 2020 and February 2022. Inclusion criteria were diagnosis of T1D for at least one year, confirmed by positivity of at least two diabetes-associated autoantibodies (GADA, ZnT8A, IAA or IA–2A), and an age between 9 and 15 years. Exclusion criteria were chronic diseases other than T1D requiring pharmacotherapy, presence of other related genetic diseases, intake of drugs that alter salivary secretion, fixed orthodontic appliances, use of antibiotics or probiotics three months prior to inclusion, use of probiotic-containing food and medical conditions believed to affect oral and gut microbiota. Written informed consent to participate in the study was obtained from the parents/guardians of the children and adolescents. The Ethical Committee of the University Hospital of Verona approved the study, in accordance with the World Medical Association Declaration of Helsinki (approval number: Prog. 2722CESC, Prot n.29192, 25/05/2020).

### 2.2. Clinical Data Collection

Clinical and demographic parameters were recorded at enrollment: age, gender, age of onset, and duration of T1D and anthropometric measurements (i.e., body height, body weight, pubertal status determined using Tanner stages I–V [34], according to standard procedures, as previously reported [34,35]. Body mass index (BMI) was calculated using the formula: body weight (kg)/body height (m^2^), values were then standardized (BMI z-score) calculating age and sex-specific BMI percentiles according to World Health Organization (WHO) child growth standards [36]. The systolic blood pressure (SBP) and diastolic blood pressure (DBP) were measured by a physician three times on the left arm with the subject sitting, using a manual sphygmomanometer and a cuff of appropriate size [37]. Other clinical data such as daily insulin dosages (total, basal) and type of treatment (multiple daily insulin injections or continuous subcutaneous insulin infusion) were also recorded. Moreover, a questionnaire on oral hygiene habits was administered the same day as the visit and saliva sample collection. The two-page self-completion questionnaire was developed following a scoping review of the literature. The following topics were covered in the questionnaire: frequency of professional and daily oral hygiene, oral health advice received, current dental care, and oral hygiene behavior.

### 2.3. Glucometabolic Parameters

Glucometabolic control parameters (glycated hemoglobin (HbA1c) and continuous glucose monitoring (CGM) metrics of glycemic control and glucose variability) were collected. HbA1c was measured using the high-performance liquid chromatography technique and standardized to the normal range established by the DCCT (4.0–6.0%, 20–42 mmol/mol). Intermittently scanned continuous glucose monitoring device (isCGM, Abbott FreeStyle Libre^®^ Glucose Monitoring System, Abbott Diabetes Care, Alameda, CA, USA) or real-time CGM device (rtCGM, Dexcom G5^®^ CGM System or Dexcom G6^®^ CGM System, Dexcom, San Diego, CA, USA)-derived data were recorded. For each participant, several metrics of glycemic control and variability have been computed separately for the full 12-week period of data collection immediately before the enrollment visit with HbA1c measurement and saliva collection. In particular, the following metrics were calculated: (a) glucose management indicator (GMI); (b) percentage of time below range [<70 mg/dL (TBR)]; (c) percentage of time in target range [70–180 mg/dL (TIR)]; (d) percentage of the time above range [>180 mg/dL (TAR)]; (e) coefficient of variation (CV). The following cut-offs were used according to the international consensus on use of CGM: TIR (using 70% as cut-off), TAR (using 25% as cut-off), CV (using 36% as cut-off) [38]. As regards HbA1c and GMI, we used a less-stringent goal (using 7.5% as cut-off) to classify patients as having good or poor glycemic control, as optimal glycemic control is often more difficult to achieve in the pediatric setting, and 7.5 represented exactly the median HbA1c of our population. Hence, an HbA1c value of <7.5% (58 mmol/mol) was taken as indicator of good glycemic control while a value of ≥7.5% (58 mmol/mol) was considered to be indicator of poor control. In order to ensure an adequate amount of data, participants were included in the analysis if at least 80% of expected CGM readings were available for each patient.

### 2.4. Microbiological Analysis

At the baseline visit a sample of 4 mL of saliva was collected while fasting for at least 8h and before performing daily oral hygiene. Samples were sent within 24 h at the Microbiology section of the Department Diagnostic and Public Health of the University of Verona and subjected to nucleic acid extraction. On saliva, bacterial culture-based analysis was also performed.

Several culture media were selected for the detection and isolation of the species of interest. In particular, Blood agar (Blood Agar Base Oxoid™) and Chocolate Agar (Chocolate Agar Base Oxoid™) have been selected as enriched media for overall oral bacterial microflora; Sabouraud agar (Oxoid™ Prepared Sabouraud Dextrose Agar) was used to isolate fungi and yeasts; Mannitol Salt agar (MSA Base Oxoid™) was employed for the growth of presumptive pathogenic staphylococci. Mitis Salivarius Agar (NutriSelect^®^ Plus) was used for the isolation of oral streptococci. Mitis salivarius sucrose bacitracin (MSB), obtained by adding 0.2 units/mL bacitracin and by increasing the sucrose concentration to 20% starting from Mitis Salivarius Agar, was employed for the selective isolation of *Streptococcus mutans*.

The plates were then incubated at 37 °C for 48 h in an anaerobic condition, except Sabouraud plates that require aerobic conditions. To detect the bacterial load (colony-forming units (CFU)/mL) a serial dilution method was performed for each saliva sample.

Nucleic acids were extracted using the QIAamp DNA Microbiome Kit (Qiagen, Milan, Italy) following the manufacturer instructions and all DNA samples were suspended in 50 μL of elution buffer. Concentrations of extracted DNAs were assessed using the Qubit 2.0 fluorometer (Invitrogen, Thermo Fisher Scientific, Darmstadt, Germany). Briefly, 10 µL of extracted genomic DNA were mixed with 190 µL of a Qubit working solution (Qubit High Sensitivity Assay, Invitrogen, Thermo Fisher Scientific, Darmstadt, Germany) according to the manufacturer’s protocol. Extracted DNA was stored at −20 °C until further use. Presence of pathogens’ DNA in the samples was assessed through PCR. Two different multiplex PCRs were performed to identify the presence of: (a) *P. gingivalis*, *P. intermedia*, and *A. actinomycetemcomitans* [39], and (b) *T. forsythia*, *T. denticola*, and *A. naeslundii*. Additionally, a single PCR was performed to identify *Actinomyces* spp., *S. mutans*, *Veillonella* spp., and *Lactobacillus* spp., as previously reported [40]. PCR reactions were performed using the 5Prime Hot Master Mix (Quantabio, Beverly, MA, USA) according to manufacturer’s instructions. Briefly, PCR reactions mixture were composed of 8 μL of 5 PRIME HotMaster Mix (2.5x), 100 nM of each primer, 50 ng of template, and Ultrapure DNase/RNase-free distilled water (Thermo Fisher Scientific, Waltham, MA, USA) to reach a final volume of 20 μL. Primers and PCR conditions are reported in Appendix A.

### 2.5. Statistical Methods

Data are presented as arithmetic mean with the relative standard deviation (SD), the medians and interquartile range [IQR], or as an absolute and relative frequency. Normal distribution of variables was assessed via the Kolmogorov–Smirnov test. Skewed variables were log-transformed unless deviations from the Gaussian distribution could not be corrected via transformation. Differences between patients stratified by sex and glycemic parameters were assessed via Student’s t-test, for Gaussian variables, and via the Mann–Whitney test, for skewed variables. A chi-square test was applied to detect differences in categorical variables. Correlations between variables were calculated by using Spearman’s rho. The relation between the presence of cariogenic bacteria (*S. mutans* and *Veillonella* spp.) and glycemic metabolic control parameters (i.e., % of HbA1c, TIR, TAR, GMI) was assessed using linear logistic regression analysis. Sex, age, bleeding during brushing, and professional hygiene frequency were used as covariates. Covariates included in multivariate regression models were selected as potential confounding factors based on their plausibility. Significance level for all tests was set at *p* < 0.05. All analyses were performed in R environment, STATA, and IBM SPSS Statistics 26 statistical package (SPSS, Chicago, IL, USA).

## 3. Results

A population of 89 children and adolescents with T1D (55 males, 61.8%) with a mean age of 12.6 ± 2.2 years was recruited. Table 1 showed the main anthropometric and metabolic characteristics and bacterial populations identified in the study sample stratified by sex and glucose control (i.e., HbA1c). Subjects’ characteristics are described according to glucometabolic control parameters (i.e., GMI, %TIR and %TAR) in Appendix A are reported.

The bacterial distribution in our population is represented in Figure 1.

*Actinomyces* spp., *A. actinomycetemcomitans*, *P. intermedia*, and *Lactobacillus* spp. were found in all investigated samples, while 93.3% of the subjects were colonized by *Veillonella* spp. *A. naeslundii*, *T. denticola*, and *T. forsythia* were identified in 47.2%, 36.0%, and 33.7% of the cohort, respectively. *S. mutans* was found in about half of the analyzed sample (49.4%). Its distribution, according to glycemic control, in terms of TIR < 70%, TAR > 25%, and HbA1c > 7.5%, showed a significantly higher percentage (all *p* < 0.03). The difference by GMI was at the limit of statistical significance (*p* = 0.052). Similarly, the concomitant presence of *Veillonella* spp. and *S. mutans*, both known as cariogenic pathogens, was higher in subjects with poor glycemic control (all glycemic metrics but GMI). No differences in CFU/mL counts were found according to the glucose control parameters (Table 1 and Appendix A).

Respondents’ use of professional dental care and oral hygiene behaviors are shown in Table 2.

Slightly less than 40% of the children and adolescents (38.7%) reported bleeding episodes, an early sign of gingivitis, during brushing. According to oral hygiene habits, only 51.4% of subjects underwent professional oral hygiene at least once a year, while approximately 65% replaced toothbrushes or brush heads every two to three months. The simultaneous presence of *T. forsythia*, *T. denticola*, and *P. gingivalis* (i.e., the red complex) negatively correlated with virtuous oral health habits such as frequency of dental visits and professional oral hygiene (rho = −0.314; *p* = 0.006 and rho = −0.263; *p* = 0.023, respectively). In addition, *S. mutans* correlates with the above poor oral hygiene practices as supported by molecular and culture-based analysis (rho = −0.280; *p* = 0.021 and rho = −0. −0.301; *p* = 0.01, respectively).

The regression analysis confirmed that *S. mutans* and *Veillonella* spp. were associated with poor glycemic control: the combination of these two bacteria was associated with higher HbA1c (OR = 3.83, 95%CI (1.26;11.65)), higher TAR (OR = 6.48, 95% CI (1.6;26.2)), lower TIR (OR = 0.21, 95% CI (0.06;0.71)), and higher GMI (OR = 4.53, 95% CI (1.25;15.15)), adjusting for age, sex, bleeding during brushing, and professional hygiene frequency as covariates (all *p* < 0.02) (Table 3).

## 4. Discussion

Our study, in accordance with the literature [10,11], concurs in emphasizing the high prevalence of cariogenic and periodontopathogenic in children and adolescents with T1D and that their oral microbiota is characterized by the presence of *Actinomyces* spp., *Lactobacillus* spp., *A. actinomycetemcomitans*, and *P. intermedia*. In addition, in a high percentage of subjects, other opportunistic bacterial species associated with dental and periodontal disease have been found, including *Veillonella* spp., *S. mutans*, *A. naeslundii*, *T. denticola*, and *T. forsythia*, indicating evident oral microbiota dysbiosis. As regards pathogens belonging to the red complex (*T. forsythia*, *T. denticola* and *P. gingivalis*), known to be associated with chronic and severe PD, 80% of the subjects had at least one of them and 30% showed the co-presence of *T. forsythia* and *T. denticola*, despite the young age of the participants [41]. Similar frequencies were found in a study on adult patients with T1D and PD in which the combination of these two periodontal pathogens correlated with poor glycemic control [42,43]. Nonetheless, in our study, levels of these periodontopathogens did not differ according to glycemic control. However, the simultaneous presence of red complex periodontopathogens negatively correlated with virtuous oral hygiene habits such as frequency of toothbrush changes and professional oral hygiene, pointing out that these modifiable factors are fundamental determinants in the prevention of dysbiosis associated with the risk of PD [44].

The detection of *A. actinomycetemcomitans* in whole analyzed samples is in accordance with previous studies reporting a high presence of this microorganism in subjects with both PD and T1D or T2D [45,46,47]. Evidence clearly underlines its etiological role in localized aggressive juvenile PD [48].

*S. mutans*, along with colonization by *Lactobacillus*, is considered one of the key elements in the early development of caries and an important factor in the predisposition of future caries risk [49]. *S. mutans* leads to an alteration of the local environment by forming an acid and exopolysaccharide-rich milieu, thus creating a favorable niche for the growth of other acidogenic and aciduric species [50]. In a study by Gross et al., the presence of *Veillonella* spp. was associated with the future development of caries in a population of healthy children, suggesting a key role of this bacteria as an early indicator of a caries-predisposing oral environment [51]. In our study, *Lactobacillus* and *Veillonella* spp. have been found in 100% and 93.3% of subjects enrolled, respectively, suggesting that an early alteration in the composition of oral microbiota could increase the risk of developing caries in children and adolescents with T1D. Despite some controversial studies [21,22,52], most recent evidence supported the presence of higher levels of cariogenic bacteria (i.e., *S. mutans*, *Veillonella* spp. and *Lactobacillus*) in patients with diabetes, particularly in subjects with poor glycemic control, supporting an association between poor glycaemic control and dysbiosis status of the oral microbiota, which is associated with a higher risk of oral and dental diseases [21,22,53,54].

Accordingly, in our study children and adolescents with poor glycemic control had a significantly higher presence of cariogenic bacteria (i.e., *S. mutans*) than peers with a good metabolic control. Presence of *S. mutans* and *Veillonella* spp was significantly associated with indicators of suboptimal glycemic control, such as higher HbA1c and TAR, and lower TIR. Although our results support the idea that an adequate metabolic control may decelerate the proliferation of pathogenic oral bacteria, it remains unclear whether the association is causative or reactive and whether an intervention to manipulate the oral microbiota could be of clinical utility for improving diabetes control [55].

The positive correlation between diabetes and pro-cariogenic bacteria may also be related to environmental factors and behavioral aspects, particularly diet. Subjects with T1D have generally higher frequency of food intake than non-diabetic peers and more frequent use of simple and complex sugar intake to deal with hypoglycaemia episodes [56]. A diet rich in fermentable carbohydrates exposes these patients to prolonged acidic conditions. Furthermore, subjects with T1D have reduced salivary flow, increased viscosity, and reduced buffering and antimicrobial capacity of saliva [57,58]. All these factors favor the selection and proliferation of acidogenic and acid-tolerant bacterial species responsible for caries development. Brushing habits and frequent dental visits are considered the main methods to prevent oral diseases as early as during childhood, including gingivitis and dental caries. Our results indicate that the habit of brushing teeth after hypoglycaemia correction with simple sugars is infrequent (22%) and, in any case, this never occurs after overnight hypoglycaemia corrections.

Our study could provide significant insights into the interaction between glycemic control, oral hygiene habits, and oral microbiota composition in subjects with T1D. Although the underlying mechanism and the mutual interaction between oral microbiota composition and diabetes and glycaemic control need to be further investigated, the presence of periodontopathogens and cariogenic bacteria is certainly an early indicator of dental and periodontal disease risk that is associated with glycemic control and oral hygiene habits. Thus, oral health education and early diagnosis and treatment of PD should be recommended to T1D subjects as early as during childhood. In this regard, young patients with diabetes and their families should be educated on proper oral hygiene care to counteract the accumulation of bacterial biofilm for the prevention of caries and PD, also given the long-term impact on metabolic control. Appropriate patient education in hygiene maintenance combined with professional oral hygiene sessions with specific devices could positively influence the oral health of subjects, and in the case of T1D subjects, also have positive impacts on metabolic health in the short and long term. Subjecting patients to a monitoring protocol and professional oral hygiene sessions could be the key to the success in preventing oral complications [30]. Patients should be monitored regularly to assess the dental and periodontal health status, and sessions of professional oral hygiene should be scheduled periodically, particularly in patients with a tendency to accumulate plaque and tartar and with early signs of PD, such as gingivitis [59,60]. Probiotics and antibacterial substances for local and/or systemic use have been the subject of intense recent investigations [61] and may provide clinical benefits in the nonsurgical treatment of periodontal disease and prevention of complications, although these systems should be further studied and analyzed [62]. The use of paraprobiotics formulations resulted in a significant reduction in most clinical indices evaluated in comparison with conventional chlorhexidine treatments in adult patients with PD and significantly reduced, after 6 months of use, the percentage of pathological bacteria, including the “red complex” ones [63]. Their immunomodulatory role and their ability to maintain or restore the balance of the oral flora appear to be promising in the prevention and therapy of oral dysbiosis and may have long-term effects not only on the course of oral pathology, but also on metabolic control and the risk of complications.

The limitations of this study include: (1) the small sample size and qualitative microbial analysis; (2) the absence of healthy controls to compare prevalence of periodontopathogens; (3) the cross-sectional study design that is not suitable for inferring causality; and (4) the absence of oral health evaluation by a dental hygienist.

The strengths of our study are: (1) the young age of the sample which, lacking evident disease processes, allows for the detection of early changes in the composition of the oral microbiota and identification of potential targets for prevention and intervention; and (2) the use of CGM data recorded for a 12-week period that allows a detailed assessment of glucose metabolism.

## 5. Conclusions

Our study shows that children and adolescents with T1D have a characteristic composition of the oral microbiota with a high prevalence of cariogenic and periodontopathogens bacteria from an early age. The presence of cariogenic bacteria and periodontopathogens are associated with glycemic control parameters and oral hygiene habits. Therefore, it is crucial to pay attention to glycemic control and regular daily and professional oral hygiene to prevent the establishment of an oral microbiota predisposing to dental and periodontal pathology in subjects with T1D since a pediatric age. Given the bidirectional relationships between oral and metabolic health, prevention and treatment of dental and periodontal health should be part of the multidisciplinary care of T1D.

## Figures and Tables

**Figure 1 microorganisms-11-00668-f001:**
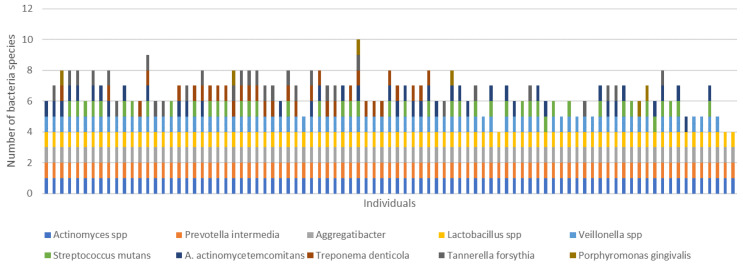
Presence of bacterial species for each individual.

**Table 1 microorganisms-11-00668-t001:** Main clinical, biochemical, and microbiota characteristics of children and adolescents with type 1 diabetes according to sex and glucose control (HbA1c, cut-off = 7.5%).

	All (*n* = 89)	Males (*n* = 55)	Females (*n* = 34)	*p*-Value	HbA1c ≤ 7.5% (*n* = 47)	HbA1c > 7.5% (*n* = 42)	*p*-Value
Female (%)	34 (38.2)				19 (40.4)	15 (35.7)	0.781
Male (%)	55 (61.8)				28 (59.6)	27 (64.3)	
Age (years)	12.56 ± 2.17	12.60 ± 2.18	12.50 ± 2.17	0.824	12.55 ± 2.11	12.58 ± 2.24	0.986
Diabetes duration (years)	6.22 ± 2.97	6.08 ± 2.82	6.43 ± 3.23	0.596	6.26 ± 3.00	6.17 ± 2.97	0.736
BMI (kg/m^2^)	20.12 ± 3.00	19.68 ± 3.03	20.82 ± 2.83	0.080	19.85 ± 3.04	20.41 ± 2.94	0.342
HbA1c (%)	7.54 ± 0.84	7.52 ± 0.87	7.57 ± 0.81	0.813			
GMI (%)	7.42 ± 0.71	7.42 ± 0.73	7.41 ± 0.68	0.911	7.04 ± 0.46	7.87 ± 0.69	<0.001
Pubertal status, *n* (%)				0.003			0.697
Prepubertal	21 (23.6)	15 (27.3)	4 (11.8)		10 (21.3)	9 (21.4)	
Pubertal	44 (49.4)	32 (58.2)	14 (41.2)		26 (55.3)	20 (47.6)	
Post-pubertal	24 (27.0)	8 (14.5)	16 (47.1)		11 (23.4)	13 (31.0)	
Total Insulin (U/kg/die)	0.88 ± 0.25	0.84 ± 0.23	0.94 ± 0.27	0.064	0.88 ± 0.22	0.88 ± 0.27	0.818
Basal Insulin (U/kg/die)	0.43 ± 0.15	0.41 ± 0.15	0.47 ± 0.14	0.064	0.44 ± 0.15	0.42 ± 0.14	0.696
Prandial Insulin (U/kg/die)	0.40 [0.29–0.55]	0.38 [0.28–0.54]	0.43 [0.33–0.58]	0.293	0.38 [0.28–0.54]	0.42 [0.33–0.56]	0.479
Time below range (%)	3.90 ± 3.61	4.28 ± 3.75	3.27 ± 3.31	0.177	4.04 ± 3.69	3.73 ± 3.55	0.690
Time in range (%)	56.64 ± 15.15	55.96 ± 14.87	57.76 ± 15.76	0.595	65.83 ± 11.03	46.34 ± 12.28	<0.001
Time above range (%)	39.40 ± 16.22	39.48 ± 16.04	39.27 ± 16.75	0.954	29.91 ± 11.96	50.05 ± 13.60	<0.001
Mean glycemia (sensor)	172.1 ± 30.0	171.9 ± 30.5	172.5 ± 29.5	0.922	156.1 ± 19.1	191.5 ± 29.4	<0.001
CV (%)	37.89 ± 5.49	38.64 ± 5.69	36.59 ± 4.95	0.104	36.65 ± 4.72	39.48 ± 6.04	0.020
Blood agar, *n* = 82 (CFU × 10^8^/mL)	10.4 ± 43.8	4.47 ± 4.17	19.7 ± 69.6	0.005	6.57 ± 5.42	14.9 ± 64.3	0.161
Sabouraud agar, *n* = 81 (CFU × 10^4^/mL)	0.35 ± 1.59	0.35 ± 1.49	0.36 ± 3.62	0.347	0.56 ± 2.13	0.13 ± 0.45	0.890
Mannitol Salt agar, *n* = 82 (CFU × 10^8^/mL)	0.10 ±0.89	1.84 ± 1.15	0.02 ± 0.035	0.133	0.19 ± 0.12	0.026 ± 0.16	0.860
Mitis Salivarius Agar, *n* = 82 (CFU × 10^7^/mL)	10.2 ± 21.5	7.59 ± 14.2	14.3 ± 29.1	0.034	11.9 ± 25.6	8.28 ± 1.55	0.262
MSB, N = 82 (CFU × 10^7^/mL)	1.36 ± 3.24	0.78 ± 1.58	2.22 ± 4.65	0.092	1.87 ± 4.16	0.82 ± 1.75	0.450
*Veillonella* spp., *n* (%)	83 (93.3)	51 (92.7)	32 (94.1)	0.583	42 (89.4)	41 (97.6)	0.129
*Actinomyces* spp., *n* (%)	89 (100)	55 (100)	34 (100)	1	47 (100)	42 (100)	1
*Actinomyces naeslundii*, *n* (%)	42 (47.2)	29 (52.7)	13 (38.2)	0.133	22 (46.8)	20 (47.6)	0.554
*Treponema denticola*, *n* (%)	32 (36.0)	17 (30.9)	15 (44.1)	0.151	19 (40.4)	13 (31.0)	0.240
*A. actinomycetemcomitans*, *n* (%)	89 (100)	55 (100)	34 (100)	1	47 (100)	42 (100)	1
*Prevotella intermedia*, *n* (%)	89 (100)	55 (100)	34 (100)	1	47 (100)	42 (100)	1
*Porphyromonas gingivalis*, *n* (%)	6 (6.7)	2 (3.6)	4 (11.8)	0.147	3 (6.4)	3 (7.1)	0.606
*Tannerella forsythia*, *n* (%)	30 (33.7)	19 (34.5)	11 (32.4)	0.510	14 (29.8)	13 (38.1)	0.273
*Streptococcus mutans*, *n* (%)	44 (49.4)	31 (56.4)	13 (38.2)	0.074	19 (40.4)	25 (59.5)	0.031
*Lactobacillus* spp., *n* (%)	89 (100)	55 (100)	34 (100)	1	47 (100)	42 (100)	1
*Veillonella spp+ Streptococcus mutans., n* (%)	42 (47.2)	29 (52.7)	13 (38.2)	0.133	17 (36.2)	25 (59.5)	0.028
*T. forsythia + T. denticola + P. gingivalis* (at least 2 of them), *n*(%)	16 (17.9)	10 (18.1)	6 (17.6)	0.592	8 (17.0)	8 (19.1)	0.510

Sample size, *n* = 89, unless otherwise indicated. Data are expressed as means ± SD, medians and interquartile range [IQR], or proportion (%). Differences between the two groups of individuals were tested using the unpaired Student’s for normally distributed variables, the Mann–Whitney U-test for non-normally distributed variables, or the chi-squared test for categorical variables, respectively. Abbreviations: BMI, Body mass index; CFU, colony-forming units; CV, coefficient of variation; GMI, glucose management indicator; MSB, mitis salivarius sucrose bacitracin.

**Table 2 microorganisms-11-00668-t002:** Oral hygiene habits questionnaire.

Questions		Frequency	Percent
1. Dentistry frequency	Never	1/74	1.35
	Only if necessary	29/74	39.19
	Once a year	16/74	21.62
	Twice a year	20/74	27.03
	More frequently	8/74	10.81
2. Professional hygiene frequency	Never	12/74	16.22
	Only if necessary	24/74	32.43
	Once a year	22/74	29.73
	Twice a year	16/74	21.62
3. Brushing frequency	Once a day	14/75	18.67
	Twice a day	46/75	61.33
	More often	15/75	20.00
4. Brushing time	Less than 1 min	15/73	20.55
	From 1 to 2 min	38/73	52.05
	More than 2 min	20/73	27.40
5. If I see blood when I brush my teeth	Normal	4/75	5.33
	It rarely happens	25/75	33.33
	It never happens	46/75	61.33
6. Brushing technique	I don’t know	16/69	23.19
	Horizontal movements	16/69	18.84
	Horizontal and vertical movements	97/69	53.62
	Others	3/69	4.35
7. Types of toothbrushes	Manual	37/75	49.33
	Electric	38/75	50.67
8. Frequency of toothbrush/toothbrush head replace	2–3 months	47/72	65.28
	6 months	7/72	9.72
	1 year	2/72	2.78
	Until is not working	16/72	22.22
9. Dental floss use	No	61/74	82.43
	Yes	13/74	17.57
10. Dental brush use	No	70/74	94.54
	Yes	4/74	5.41
11. Mouthwash use	No	43/74	58.11
	Yes	31/74	41.89
12. Do you brush your teeth when correcting hypoglycemia	Never	57/73	78.08
	Sometimes	15/73	20.55
	Always during the day	1/73	1.37

**Table 3 microorganisms-11-00668-t003:** Binary logistic regression analysis, where presence of *Streptococcus mutans* and *Veillonella* spp. is the response variable. The estimates were adjusted for gender, age class, professional hygiene frequency, and bleeding during brushing.

Variable		OR	*p*-Value	[95% CI]
Dependent	Independent			
*Streptococcus mutans* and *Veionella* spp.	HbA1c	3.83	0.018	1.26;11.65
	Gender	0.40	0.080	0.14;1.12
	Age	0.97	0.952	0.35;2.69
	Professional hygiene frequency	1.10	0.865	0.35;3.48
	Bleeding during brushing	1.56	0.407	0.55;4.45
	Pseudo R2 = 0.0969			
	GMI	4.35	0.021	1.25;15.15
	Gender	0.42	0.101	0.15;1.19
	Age	1.29	0.631	0.46;3.63
	Professional hygiene frequency	0.89	0.835	0.31;2.58
	Bleeding during brushing	1.86	0.254	0.64;5.40
	Pseudo R2 = 0.0883			
	TAR	6.48	0.009	1.60;26.20
	Gender	0.32	0.040	0.11;0.95
	Age	1.55	0.416	0.54;4.44
	Professional hygiene frequency	0.84	0.750	0.29;2.44
	Bleeding during brushing	1.79	0.291	0.61;5.24
	Pseudo R2 = 0.1153			
	TIR	0.15	0.007	0.04;0.60
	Gender	0.33	0.048	0.11;0.99
	Age	1.50	0.454	0.52;4.30
	Professional hygiene frequency	0.87	0.798	0.30;2.53
	Bleeding during brushing	1.75	0.309	0.60;5.12
	Pseudo R2 = 0.1212			

## Data Availability

All data generated or analyzed during this study are included in this article. Further enquiries can be directed to the corresponding author.

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
