# Peer review of "Oral Microbiota in Children and Adolescents with Type 1 Diabetes Mellitus: Novel Insights into the Pathogenesis of Dental and Periodontal Disease"

_microorganisms, 2023, doi:10.3390/microorganisms11030668_

Round 1
Reviewer 1 Report
Manuscript of considerable interest for the dental sector, needs a major revision.
Abstract, to better highlight the results obtained.
Keywords, few and not present on MeSH, add more
Introduction: add the reference on the new classification of periodontal disease combined with diabetic pathology, and how the oral microbiota changes according to the various age groups
Materials and methods: poorly described, expand them
Very confusing results, reorganize them by highlighting the results obtained
Discussion: add as objectives the proactive action through the use of paraprobiotics and probiotics as already studied by the research group of Prof Scribante.
Conclusions: rephrase them based on the comments
Bibliography: add references required
Reviewer 2 Report
GIVEN THE HIGHLIGHTED CORRELATION BETWEEN presence of diabete AND PERIODONTAL DISEASE, IT IS IMPORTANT TO FURTHER EMPHASISE THAT ORAL HYGIENE FOLLOW-UPS ARE CRUCIAL FOR THE DEFENCE OF the patient to the DISEASe DESCRIBED.
Introduction and discussion should be improved with the suggested references
Materials and methods are well described and pertinent
CONCLUSION is correct and interesting.
AUTHORS SHOULD PLACE MORE EMPHASIS ON THE IMPORTANCE OF HYGIENE FOLLOW-UPS USING THE RECOMMENDED BIBLIOGRAPHIES
PMID: PubMed ID 34425662
PMID: PubMed ID 34425659
PMID: PubMed ID 34425666
Reviewer 3 Report
Comments to Authors
In this manuscript, authors investigated the interplay between bacteria composition, oral hygiene, and glycemic control in a cohort of children with type 1 diabetes mellitus (T1D). They found that subjects with poor glycemic control (HbA1c > 7.5%) had a higher percentage of Streptococcus mutans and Veillonella spp. (OR = 3.83, 95%CI (1.26;11.65)). These findings were fundamentally interesting for understanding the relationship between glycemic control and oral hygiene to prevent oral diseases. However, this manuscript has some problems.
Major points
1. Line 63-64. Previous studies showed that the simultaneous presence of Veillonella spp. and Streptococcus spp. can promote the development and progression of dental caries. Line 278-283. The authors mentioned, “Our study could provide significant insights into the interaction between glycemic control, oral hygiene habits, and oral microbiota composition in subjects with T1D. Although the underlying mechanism and the mutual interaction between oral microbiota composition and diabetes and glycemic control need to be further investigated, the presence of periodontopathogens and cariogenic bacteria is certainly an early indicator of dental and PD risk associated with glycemic control and oral hygiene habits”. But, they only showed that S. mutans + Veiollonella spp. was highly detected in patients with high HbA1c. The presence of periodontal pathogens, including red complex, was not different between high and low HbA1c patients. Why do they think these pathogens can be an indicator of PD risk?
Minor points
1. Line 247. “Lattobacillus” is a misspelling.
Reviewer 4 Report
Authors provide a study to evaluate the association between the bacteria composition, oral hygiene, and glycemic control in a cohort with Type1 Diabetes. It demonstrates a negative correlation between the red complex bacteria and the oral hygiene habits. However, several concerns need to be solved before the work is considered for publication.
1. A major limitation of the study is the lack of dental evaluation. As the main finding is about the periodontal pathogens. Although the oral health is not evaluated by a dental hygienist, the author shall provide a way to show that the observation is not a result of confounding with dental disease.
2. The result shows a negative correlation between the co-occurrence of all three red complex species and oral health habits, which is considered as a main finding of the study. How many individuals demonstrate simultaneous presence of the three red complex bacteria? According Fig. 1, the three species simultaneously present in only two individuals. Thus, the conclusion is not very convincing.
3. How were the cut-off of HbA1C and so on determined? Do they have clinical meaning?
4. Authors claimed that the subjects with poor glycemic control had higher percentage of S. mutans and Veillonella. spp. However, Veillonella. spp colonized in 93.3% subjects, which are almost all the subjects enrolled in the study. What if they were analyzed independently? Actinomyces spp. was also considered as a carry contributor. Why Actinomyces spp. was not included in this analysis, which is designed to explore the association between cariogenic bacteria and glycemic control? If this is not tested for cariogenic bacteria, the analysis design needs to be justified.
Round 2
Reviewer 1 Report
The Manuscript has been properly reviewed according to the comments of the first round of review, it can be published
Author Response
We would like to thank the reviewer.
Reviewer 4 Report
Thanks for author's efforts to solve the concerns. However, there are still some minor concerns shall be clarified.
1. I cannot agree with the authors response regarding the analysis of Veillonella spp and S. mutans. The following is part of author's response:
The presence of both Veillonella spp and S. mutans was independently evaluated, as shown in Table 1, according to gender and glycemic control. However, only S. mutans was singularly statistically associated with glycemic control: the higher the HbA1c the greater the presence of bacteria.
The authors response supports my point that the significant association between the presence of both Veillonella spp and S. mutans and poor glycemic control could be a result of that S. mutans demonstrates significantly association while Veillonella spp presents in almost all the samples. Current evidence is not enough to support the conclusion. It is more likely an overstatement. A multivariate analysis including the intersection term of Veillonella spp and S. mutans could help to clarify the point.
2. What about the combination of Veillonella spp and red complex bacteria, especially P. gingivalis? Literatures have shown that the early colonization of Veillonella spp may help the colonization of P. gingivalis and increase the risk of periodontitis.
